# GENETIC ALGORITHM FOR CONSTRAINED MOLECULAR INVERSE DESIGN

## ABSTRACT

A genetic algorithm is suitable for exploring large search spaces as it finds an approximate solution. Because of this advantage, genetic algorithm is effective in exploring vast and unknown space such as molecular search space. Though the algorithm is suitable for searching vast chemical space, it is difficult to optimize pharmacological properties while maintaining molecular substructure. To solve this issue, we introduce a genetic algorithm featuring a constrained molecular inverse design. The proposed algorithm successfully produces valid molecules for crossover and mutation. Furthermore, it optimizes specific properties while adhering to structural constraints using a two-phase optimization. Experiments prove that our algorithm effectively finds molecules that satisfy specific properties while maintaining structural constraints.

## 1 INTRODUCTION

When the search space is complex or partially known, it is difficult to optimize current solutions using gradient descent (Jin & Ha, 1997; Ahmad et al., 2010). Because genetic algorithms find approximate solutions, they are effective in exploring vast unknown space such as molecular search space. (Holland, 1992; Jin & Ha, 1997; Ahmad et al., 2010; Henault et al., 2020). In drug discovery, optimizing complex pharmacological properties using genetic algorithms have been widely studied and extended (Leardi, 2001).

Deep learning-based molecular design is being actively conducted to improve pharmacological properties and has served as a powerful tool (Gómez-Bombarelli et al., 2018). However, inferred drug candidates depending on the architecture did not satisfy the heuristic guidelines set by chemical researchers (George & Hautier, 2020). Furthermore, this not only produces non-synthesizable molecules but also causes optimization problems for non-linear structure-activity relationships (Gómez-Bombarelli et al., 2018; Vanhaelen et al., 2020).

Most molecular design methods first generate molecular structures and then calculate properties of the structures resulting in high computational cost (Duvenaud et al., 2015; Gilmer et al., 2017; Feinberg et al., 2018; Yang et al., 2019). Conversely, inverse molecular design specifies target properties in advance and then systematically explores the chemical space to discover molecular structures that have the desired properties (Sanchez-Lengeling & Aspuru-Guzik, 2018).

Recent advances in molecular inverse design of genetic algorithms compete with or even surpass deep learning-based methods (Yoshikawa et al., 2018; Jensen, 2019; Nigam et al., 2019; Polishchuk, 2020; Ahn et al., 2020; Nigam et al., 2021b;a). This indicates that the computational methods of genetic algorithms in a chemical domain are more effective for exploring the vast chemical space (Ahn et al., 2020). However, the genetic operators cause difficulty in lead optimization where the molecular structure is constrained (Hasançebi & Erbatur, 2000).

A common molecular design strategy is to narrow the chemical search space, starting with known potential molecules (Lim et al., 2020). The scaffold, which is the "core" of the molecule is intentionally maintained to preserve basic bioactivity (Hu et al., 2016). This is directly involved in the interaction with the target protein (Zhao & Dietrich, 2015; Lim et al., 2020). In lead identification process, scaffolds with biological activity to the target protein are identified (Zhao & Dietrich, 2015). In lead optimization process, it is important to optimize the SAR(Structure-Activity Relationship) properties while staying in the chemical space associated with the privileged scaffold (Langevin

et al., 2020). Lead optimization can be described as a multiple optimization problem in scaffold constraints (Langevin et al., 2020). Most studies have focused on the application of generative models to the field of medicinal chemistry, and studies related to structure-constrained lead optimization have not been extensively explored.

In this regard, we introduce a genetic algorithm featuring constrained molecular inverse design. We use graph and SELFIES descriptors to generate valid molecules through a two-phase GA-based pipeline. In addition, we introduced a two-phase optimization to ensure that molecular generation does not fail to optimize for specific properties while adhering to structural constraints.

## 2  RELATED WORK

Among the various methods for constrained optimization in a genetic algorithm, the basic one is designing effective penalty functions (Yeniay, 2005). The functions, which impose a penalty to fitness value, are widely used for constrained optimization (Yeniay, 2005; Fletcher, 2013). The individual fitness value is determined by combining the objective function value with the constraint violation penalty. A dominant relationship exists between the constraint penalty and the objective function value (Coello, 2000).

One of the various methods is focusing on the selection of practicable solutions. To solve the problem of constrained optimization using a genetic algorithm, a two-phase framework which is Multi-Objective Evolutionary Algorithm (MOEA) was introduced (Venkatraman & Yen, 2005). In the first phase, MoEA confirms the constraint satisfaction of solutions using a penalty function. The algorithm ranks solutions based on violations of constraints, completely disregarding objective function. When one or more feasible solutions are identified, the second phase continues. In the second phase, individual fitness is reassigned according to the objective function-constraint violation space and bi-objective optimization is performed (Venkatraman & Yen, 2005).

Structure-constrained molecular optimization work was first presented in the JT-VAE (Jin et al., 2018a) study. In this study, they create a tree-structured scaffold for constraining the structure of a molecule and then generate a molecular graph using a message-passing network (Jin et al., 2018a). GCPN (You et al., 2018) uses goal-directed reinforcement learning to generate molecules with desired properties and similarities. VJTNN (Jin et al., 2018b) treated molecular optimization as a graph-to-graph translation problem and solved it through MMPA(Matched Molecular Pair Analysis) (Dalke et al., 2018) dataset. DEFactor (Assouel et al., 2018) generated molecules with optimized penalized LogP while maintaining molecular similarity through differentiable conditional probability-based graph VAE.

Glen & Payne proposed a method for generating molecules through domain-specific rule-based crossover and mutation to ensure molecular structure constraints (Glen & Payne, 1995). To produce offspring molecules with the parent's substructures, the crossover is used as a strategy to cut the end part of the molecule and link it to the end part of another similar molecule. To produce new molecules, twelve operators of mutation were defined: atomic insertion, atomic deletion, etc (Glen & Payne, 1995).

ChemGE (Yoshikawa et al., 2018) uses the grammatical evolution of SMILES (Simplified Molecular-Input Line-Entry System) (Weininger, 1988) to optimize penalized LogP and KITH protein inhibitors. Their study converts SMILES into integer sequences using a chromosomal mapping process (Yoshikawa et al., 2018). Subsequently, operators of mutation are used to optimize grammatical evolutionary molecular populations. GA-GB (Jensen, 2019) and MolFinder (Kwon & Lee, 2021) defined expert rules for operators of crossover and mutation to ensure the validity of the structure when generating molecules. GEGL (Ahn et al., 2020) created optimized SMILES strings for penalized LogP by presenting a reinforcement learning contained expert policy.

Various studies have proposed SMILES-based approaches since they are convenient to convert a complex 3D chemical structure to a simple 1D string (Yoshikawa et al., 2018; Kwon & Lee, 2021; Ahn et al., 2020). However, adding atoms or square brackets to a string indeed changes the structure globally not locally (Dalke, 2018). It is difficult to generate a completely valid molecule because SMILES are context sensitive (Dalke, 2018; Kwon & Lee, 2021). Recently, the development and application of SELFIES(Self-Referencing Embedded Strings), which is a 100% valid string representation, has been implemented in the molecular inverse design to cope with this problem (Krenn

et al., 2020). The GA-D (Nigam et al., 2019) generated SELFIES strings using a random operator of mutation while maintaining validity. Furthermore, this study maintained the molecular diversity of the population through an adaptive penalty of the deep neural discriminator (Nigam et al., 2019). The STONED (Nigam et al., 2021b) study was introduced for performing local chemical space search and molecular interpolation using SELFIES. Furthermore, the mutation site was limited to terminal 10% to maintain the molecular scaffold (Nigam et al., 2021b). SELFIES makes it possible to generate new molecules through the random operation without relying on expert rules in operations of mutation (Nigam et al., 2021b).

## 3 PROPOSED METHOD

In this section, we introduce a novel genetic algorithm for constrained optimization in molecular inverse design. Our algorithm generates molecules suitable for the target properties while constraining the structural similarity of a target molecule. We use two strategies to satisfy the constraint conditions. First, our algorithm constructs a population that always satisfies the structural similarity condition in the first phase. Second, the algorithm selects the appropriate molecular descriptors according to the genetic operators to ensure the validity of the molecule. The detailed whole process is shown in figure 1.

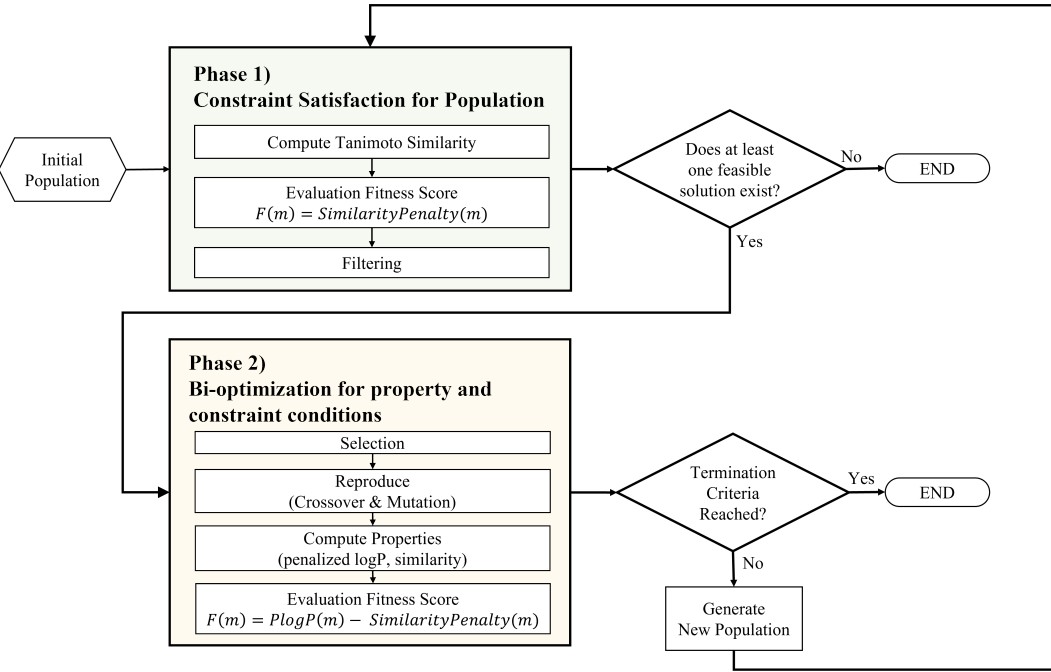

Figure 1: Illustration of genetic algorithm process for constrained optimization

The algorithm starts by constructing a population depending on whether the target molecule or similar molecules exist in the dataset. In the case of the target molecule or its similar molecule existing in the dataset, the population consists of molecules above the similarity calculated by the similarity between the molecules in the dataset and the target molecule. On the contrary, a target molecular SMILES is randomly arranged to construct a population according to the population size.

Next, our algorithm confirms constraint satisfaction from the population and proceeds with bi-optimization for property and constraint conditions. The detailed process is explained in the subsections below.

### 3.1 TWO-PHASE MOLECULAR OPTIMIZATION

In the work of constrained molecular optimization, the optimization often failed when a large number of molecules were subject to penalty included in the population. Therefore, we focused on ensuring that the population always contains only feasible solutions. Inspired by MOEA (Venkatraman & Yen, 2005), we divided the process into constraint satisfaction and bi-optimization, each of which is further explained in the following.

**Constraint Satisfaction for Population**  In this phase, the algorithm searches for feasible solutions considering only the structural similarity. The objective function is not considered in this process. The algorithm measures the Tanimoto coefficient of how similar it is to the reference molecule(Nigam et al., 2019). The similarity between the target molecule m and the product molecule $m'$ is expressed as $sim(m, m')$. $SimilarityPenalty(m)$ is given to each molecule according to as follows:

$$SimilarityPenalty(m) = \begin{cases} 0, sim(m, m') \geq \delta \\ -10^6, sim(m, m') < \delta \end{cases} \tag{1}$$

$SimilarityPenalty(m)$ is set to 0 if $sim(m, m') \geq \delta$ (Nigam et al., 2019). Otherwise, a death penalty of $-10^6$ is given. Next, the algorithm assigns fitness value to each individual by equation 2 and selects only feasible solutions. A fitness function is expressed as follows:

$$F(m) = SimilarityPenalty(m) \tag{2}$$

**Bi-Optimization for Property and Constraint Condition**  In this phase, the algorithm switches to bi-optimization for the fitness function value and the constraint condition when one or more feasible solutions are found. Our work simultaneously is to maximize the penalized $LogP$ value while retaining the molecular structure. The penalized $LogP$ ($J(m)$) of each molecule is expressed as follows:

$$J(m) = LopP(m) - SAScore(m) - RingPenalty(m) \tag{3}$$

The higher score for $J(m)$ has a more suitable structural profile as a drug (Yoshikawa et al., 2018). $LogP(m)$ is the octanol-water partition coefficient for the molecule $m$. $SAScore(m)$ is the Synthetic Accessibility score, which is a quantitative score for whether a molecular structure can be synthesized (Ertl & Schuffenhauer, 2009). The higher the score, the more difficult it is to synthesize molecules. By giving a penalty according to the score, it is possible to prevent the generation of molecules that cannot be synthesized. $RingPenalty(m)$ is used to give a penalty to prevent the creation of molecules with unrealistically many carbon rings. This function is a penalty for molecules with seven or more rings of carbon (Yoshikawa et al., 2018).

The algorithm reassigns the fitness value to each individual according to equation 4 and sorted by rank. A fitness function can be represented as follows:

$$F(m) = J(m) - SimilarityPenalty(m) \tag{4}$$

We wanted not only to preserve the superior individuals but also to replace the inferior individuals through reproduction. A portion of the generated molecules are replaced while the rest of the are kept through the selection module. The probability of replacing a molecule is determined using a smooth logistic function based on a ranking of fitness (Nigam et al., 2019).

### 3.2 GENETIC OPERATORS

In chemistry, expert rules are required to create valid molecules through the operation of crossover and mutation. We used graph molecule descriptor for crossover and SELFIES molecule descriptor for mutation. The detailed process is shown in figure 2.

**Crossover**  Performing one-point crossover on a string is faster than the operation of a graph. However, arbitrarily cutting strings does not take the molecular structure into account. We used an operator of graph-based crossover presented in GB-GA (Jensen, 2019) Considering the structure, our algorithm distinguishes between cutting and not cutting rings. There are two types of cutting methods considering the molecular structure: non-ring cut and ring cut. The probabilities are all the same in both cases.

An operation of the crossover is used to maintain a diversity of individuals. This operation first selects two-parent molecules from the population randomly. Next, it performs a ring crossover or non-ring crossover according to a set probability. The bonding of fragmented molecules proceeds according to the rules of reaction. It generates a minimum of 0 to a maximum of 4 new offspring molecules through an operation of crossover. To maintain the population, only one is randomly selected from the produced offspring molecules.

**Mutation**  An operation of mutation uses the SELFIES (Krenn et al., 2020) strings, which can produce 100% valid molecules with string conversion without special rules. The SELFIES strings can maintain the validity of the molecule even when a random point operation of mutation is performed in STONED (Nigam et al., 2021b). Therefore, we performed a one-point mutation using SELFIES which is lighter and more efficient than graph-based. There are three types of one-point mutation to maintain a diversity of individuals: atom replacement, atom insert, and atom deletion. The probabilities are all the same in all three kinds of cases. Although the SELFIES strings produce valid molecules, we wanted the mutation to act as bioisosteric replacements. We restricted the range of the operation of mutation to the terminal 10% (Nigam et al., 2021b). When a mutation is performed, the specified probability determines which operator of a mutation to perform. The position of the point is randomly determined within the specified range, and then the corresponding operation is performed.

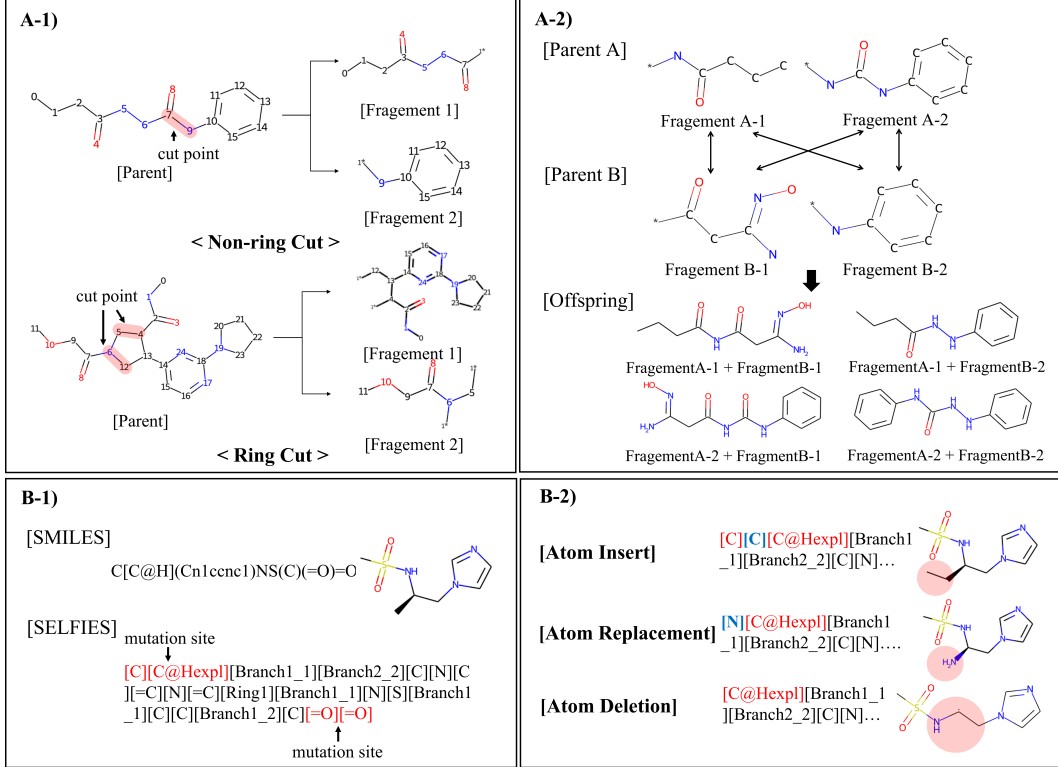

Figure 2: Illustration of genetic operations. **A-1)** Operators of crossover **A-2)** Operation of crossover **B-1)** SMILES to SELFIES transformation and mutation restriction sites(terminal 10%) **B-2)** Operators of mutation

## 4 EXPERIMENTS

### 4.1 DATA SET

The experiments used 250,000 commercially available molecules extracted from ZINC (Irwin et al., 2012). $LogP$, $SAScore$, and $RingPenalty$ constituting $J(m)$ for the molecule $m$ are normalized based on the corresponding data set. The optimization target is 800 molecules with the lowest penalized logP value in the data set.

### 4.2 SIMULATION RESULTS

In this section, we compared our model with various baseline models using penalized $LogP$. All generated molecules satisfy the constraint that the structural similarity between the original molecule and the generated molecule is higher than the fixed threshold.

This work is a realistic scenario where drug discovery usually starts from known molecules, such as existing drugs (Besnard et al., 2012; Lim et al., 2020). For comparison with other baseline models, the molecular generation size was limited to 81 SMILES lengths. This is because the value of $LogP$ varies depending on the number and type of atoms, and a good score can be obtained by increasing the molecular size to make it arbitrarily large (Jensen, 2019). $\delta$ is 0.4 or 0.6 which is the Tanimoto similarity threshold. For each molecule, the maximization was performed for 20 generations to measure the greatest. The improvement measurement results are shown in Table 1.

|  | $\delta \geq 0.4$ | | $\delta \geq 0.6$ | |
|---|---|---|---|---|
|  | Improvement | Sucess | Improvement | Sucess |
| JT-VAE | 0.84±1.45 | 84% | 0.21±0.71 | 46.4% |
| GCPN | 2.49±1.30 | 100% | 0.79±0.63 | 100% |
| MMPA | 3.29±1.12 | - | 1.65±1.44 | - |
| DEFactor | 3.41±1.67 | 85.9% | 1.55±1.19 | 72.6% |
| VJTNN | 3.55±1.67 | - | 2.33±1.17 | - |
| GA-DNN | 5.93±1.41 | 100% | 3.44±1.09 | 99.8% |
| **Constrained GA(Ours)** | **5.53±1.29** | **100%** | **3.67±1.29** | **100%** |

Table 1: Comparison on constrained improvement of penalized $LogP$ of specific molecules

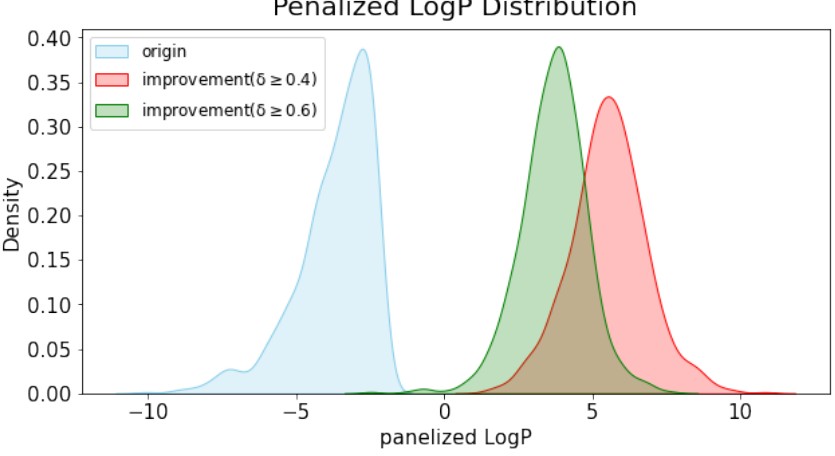

Figure 3: Distribution of penalized $LogP$

Figure 3 shows the distribution of how much penalized $LogP$ has improved compared to the original molecules. The penalized LogP of molecules was maximized, and the distribution shifted from right to left. In the case of the $\delta \geq 0.4$, it can be seen that the distribution of penalized $LogP$ is more

skewed to the right than $\delta \geq 0.6$ because the structural restriction is lower. In Figure 4, molecules with maximized penalized $LogP$ were visualized while maintaining structural constraints.

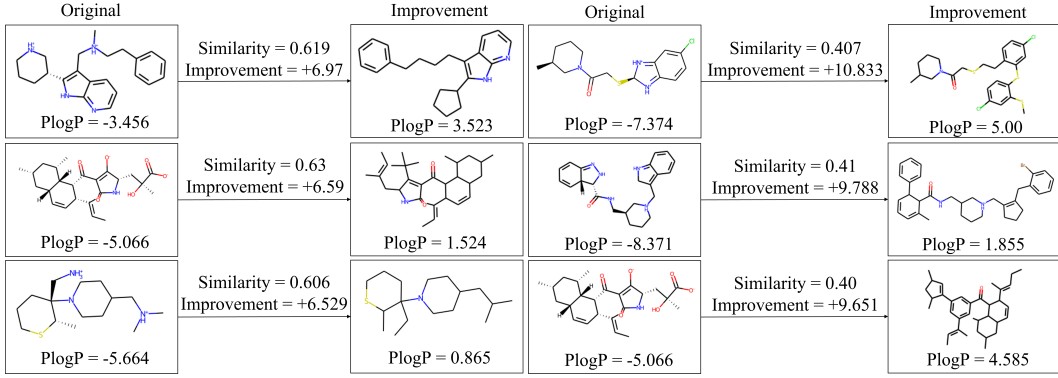

Figure 4: Compounds generated at the twentieth iteration (left : $\delta \geq 0.4$ / right : $\delta \geq 0.6$)

We focused on cannabidiol molecules which are expected to have promise for targeting various disease proteins. Cannabidiol is a natural product extracted from cannabis and is a non-psychoactive ingredient. This is attracting attention as a promising molecule that can improve Alzheimer's disease symptoms by reducing beta-amyloid accumulation and amyloid plaque prodcution, which affects the pathogenesis of Alzheimer's disease (Cooray et al., 2020; Jiang et al., 2021). We experimented with penalized LogP molecular optimization of cannabidiol. The Tanimoto similarity coefficient is set to 0.6, and optimization is carried out up to the 19th generation. The Murcko Scaffold (Bemis & Murcko, 1996) of the cannabidiol molecule is compared with the generated molecules. Duplicate molecules are removed and a total of 11 molecules are compared. The visualization result is shown in figure 5. In this figure, the yellow site is a molecular scaffold and the red sites are the altered points in the target molecule. Although some generations do not retain the scaffold, it can be seen that most of the generated molecules do retain the scaffold. Moreover, the scaffolds of all the generating molecules contain cannabidiol scaffolds as substructures. Furthermore, Hydrophobic carbon-valent functional groups were mainly added which can improve the penalized $LogP$ value.

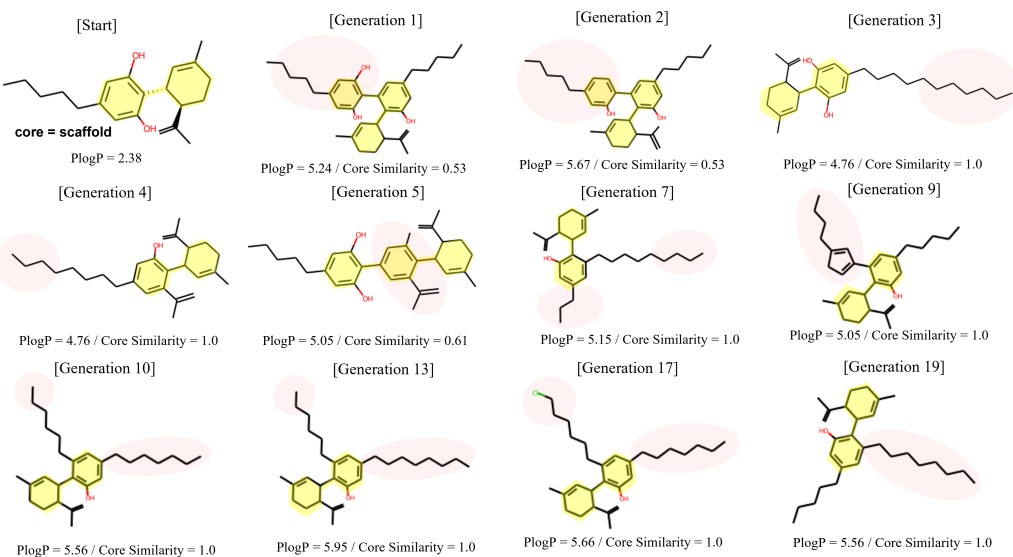

Figure 5: Optimization of Penalized $LogP$ for Synthetic Cannabidiol($\delta \geq 0.6$)

## 5 CONCLUSION

In molecular inverse design, the computational method of genetic algorithms is effective in exploring vast chemical spaces. A common molecular design strategy is to narrow the chemical search space, starting with known potential molecules. In lead optimization process, the scaffold, the "core" of the molecule, is intentionally maintained to preserve basic bioactivity. This process of optimizing SAR(Structure-Activity Relationship) properties is a multi-objective problem that maintains chemical space associated with the privileged scaffold.

In this work, we demonstrated a molecular inverse design that maximizes penalized LogP while constraining the molecular structure. The proposed method constructs a solution set in which the molecular population always satisfies the structural similarity through two-phase optimization. To generate valid molecules, we used to graph and SELFIES molecular descriptors. Our algorithm effectively generated optimized molecules while maintaining structural similarity to target molecules. Experiment shows that excels over the state of arts model. In cannabidiol molecular optimization, our model is able to maintain the molecular core and optimize target properties over generations.

It is very important to design new molecules and materials with specific structures and functions. In light of our experimental results, optimizing only penalized LogP may not take the diversity of molecular functional groups into account when binding to protein targets. In this regard, we need an extension method that can consider various properties at the same time. Our algorithm uses a docking module to enable inverse molecular design with enhanced binding affinity for biological targets. Moreover, it is expected to be used in design peptides or proteins with the desired function.

## 6 REPRODUCIBILITY STATEMENT

The genetic algorithm parameters are as follows; The population size was set at 100. The algorithm generates 1000 offspring molecules for each generation. The probability of crossover and mutation are 100% and 50% respectively. The extent of mutations restricts to the terminal 10%. The code for reproducibility is posted in the url anonymous repository: https://anonymous.4open. science/r/Reproducibility

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
