# OpenReview forum: "Genetic Algorithm for Constrained Molecular Inverse Design"
_ICLR.cc/2022/Conference — ICLR 2022 Submitted_

### Official Review · Reviewer_YxXB · 2021-11-01

**Correctness:** 3
**Technical Novelty And Significance:** 2
**Empirical Novelty And Significance:** 2
**Recommendation:** 3
**Confidence:** 4

**Main Review:**

### ===== Detailed pros: =====



1. On a high level, molecular optimization under structural constraints is an important research direction, and it has relevance to real-world tasks.

2. Results on the core optimization task look promising.


### ===== Detailed cons: =====



1. The experiments in the paper are limited and not very relevant to real-world drug discovery. Moreover, as the algorithmic novelty is rather mild, I'd expect a more thorough experimental evaluation.

(a) The only explored objective is penalized logp optimization from the JT-VAE work. While this is slightly more realistic than optimizing for logp alone, it's still a rather toy-ish optimization objective, which can be seen from Figure 5 (most improvements are based on attaching long carbon chains). It would be more interesting to explore a variety of more complex tasks, like optimizing against learned property predictors [1], Guacamol benchmarks [2], docking score optimization [3] or scaffold-constrained optimization [4,5].

(b) While the paper talks a lot about structural constraints, the way this is realized is through a hard constrained on similarity with a seed molecule. In real-world usecases, it is more common to have a subgraph (scaffold) constraint as a hard constraint, and similarity to seed as a soft constraint. Setting up the problem like this would also showcase some of the unique abilities of genetic algorithms: while it's hard to "bake in" the similarity constraint into *any* method (i.e. perform optimization without ever breaking the constraint), it is possible to define mutation/crossover operations for genetic algorithms that maintain the subgraph (scaffold) constraint. It would be interesting to see how a scaffold-constrained genetic algorithm does on scaffold-based optimization tasks [4,5].

(c) A well-known pitfall of genetic algorithms when applied to drug discovery is producing molecules that are unreasonable (toxic, unstable, etc). This should be assessed through post-filtering checks like quality filters (e.g. [6]).



2. Several parts of the paper are unclear.

(a) When the algorithm cannot find molecules that satisfy the constraint to form the initial population, the authors say the initial SMILES are "randomly arranged". Does this mean random SMILES strings are made up (e.g. by randomly adding atoms), or that random (entire) SMILES strings are randomly selected from the dataset?

(b) When describing the different evolutionary operators and the choices there, the paper says "probabilities are the same", and it's unclear what that exactly means: for example, when you have to add a new atom, is the probability of choosing each of the atom types uniform?

(c) Unclear what "we restricted the range of the operation of mutation to the terminal 10%" means.

(d) Unclear what "optimization target is 800 molecules with the lower penalized logP value in the data set" means.

(e) The conclusion says "Our algorithm uses a docking module"; I'm not sure what this means, since this is the first time docking is even brought up. I'm assuming this was meant to describe future work.



3. Prior work is somewhat misrepresented.

(a) At the top of page 2, the authors say that structure-constrained optimization has not been extensively explored, while there are many scaffold-constraints models out there [4,5,7].

(b) The authors claim JT-VAE is a *structure constrained* model due to the junction tree representation, which I don't believe is fully accurate: the junction tree is an intermediate representation, which isn't meant to constrain the *final* molecules in a specific way (as any molecule can be decomposed as a junction tree). Instead, the junction tree allows the model to keep the *intermediate* states of generation reasonable (and avoid e.g. unfinished aromatic rings). While the JT-VAE work indeed evaluates on similarity-constrained optimization, the model itself isn't specially geared towards this task, and it doesn't introduce special constraints during the generation procedure that would help to meet the similarity threshold.



### =====  Nits (didn't influence my score, just here to help): =====



- appendix: "algorithms is effective" -> "algorithms are effective"

- "The proposed algorithm successfully produces valid molecules for crossover and mutation" -> I guess replace "for" with "through"

- first paragraph of intro, "have been studied" -> "has been studied"

- second paragraph of intro, unclear what "depending on the architecture" means there

- top of page 2: "multiple optimization" -> "multi-objective optimization", "in scaffold constraints" -> "under scaffold constraints"

- bottom of page 3: "on the contrary" -> "otherwise"

- typos in figures: "fragement", "panelized"

- caption of Figure 3 says the distribution has shifted from right to left, whereas I think it's the other way around



### ===== References =====



[1] Hierarchical Generation of Molecular Graphs using Structural Motifs

[2] GuacaMol: Benchmarking Models for de Novo Molecular Design

[3] We Should at Least Be Able to Design Molecules That Dock Well

[4] Scaffold-based molecular design with a graph generative model

[5] Learning to Extend Molecular Scaffolds with Structural Motifs

[6] https://github.com/PatWalters/rd_filters

[7] Scaffold-constrained molecular generation

**Summary Of The Paper:**

This paper proposes to use a two-stage genetic algorithm for similarity-constrained molecular optimization. The authors define chemically-relevant strategies for crossover/mutation, and show good results on constrained optimization of penalized logp.

**Summary Of The Review:**

The paper explores reasonable techniques, but the experimental evaluation is limited, and the overall setting is not very relevant to real-world tasks. Moreover, many things are left unclear. I believe the paper needs more work to be made ready for publication.

---

### Official Review · Reviewer_WQyJ · 2021-11-02

**Correctness:** 2
**Technical Novelty And Significance:** 2
**Empirical Novelty And Significance:** 2
**Recommendation:** 3
**Confidence:** 4

**Main Review:**

1.	Compared to (Nigam et al., 2019), the objective function is almost the same, i.e., Eqn.(3) in this paper and Eqn.(2) in (Nigam et al., 2019). The crossover and mutation operations are also used in (Nigam et al., 2019).
2.	The authors only conduct experiments on improving LogP, which cannot show that this is a general method for molecule generation.
3.	The improvement over GA-DNN is limited.


**Summary Of The Paper:**

The authors proposed a genetic algorithm for molecule generation. The reward is a combination of several objectives like Eqn.(3). The genetic operations include crossover and mutation. The authors conduct experiments to optimize the LogP of molecules.

**Summary Of The Review:**

The method lacks novelty and experiments are not convincing.

---

### Official Review · Reviewer_QMQj · 2021-11-02

**Correctness:** 2
**Technical Novelty And Significance:** 1
**Empirical Novelty And Significance:** Not applicable
**Recommendation:** 3
**Confidence:** 5

**Main Review:**

Strength:
 * Related work section is comprehensive.

Weakness:
 * Weak and incomplete results. In Table 1, the model performance is quite similar to GA-DNN baseline. For $\delta>0.4$, constrained GA (proposed method) is 5.53 while GA-DNN achieves higher logP improvement 5.93. The original benchmark proposed by Jin et al. 2019 also did molecular optimization for QED and DRD2, but authors did not report any results on these two properties.
 * Lack of novelty. The proposed method is a direct application of standard genetic algorithm. The only difference is that this paper applies a two-stage optimization procedure that enforces the similarity constraint first and then optimize the property. The cross-over operator is the same as GB-GA (Jensen 2019), and the mutation operation is based on SELFIES string (Krenn et al. 2020).
* There is no ablation study of modeling choices. What's the benefit of two-stage procedure? Does that make a huge difference in model performance? The main claim of the paper (benefit of two-stage procedure) is not supported by empirical results.

**Summary Of The Paper:**

This paper proposes a genetic algorithm for constrained molecular optimization. To generate molecules with better properties while staying similar to the starting molecule (lead), this paper proposes a two phase procedure. The first stage (constraint satisfaction) searches for feasible molecules that are similar to the lead compound. The second stage optimizes the objective function (property). The crossover operation is the same as GB-GA (Jensen 2019) and the mutation operation introduces random modifications on SELFIES string representation. The method is evaluated on logP optimization task, with minor improvement over previous work GA-DNN (Table 1).

**Summary Of The Review:**

I vote for rejection. The method itself is not novel because it is a standard application of genetic algorithm. The results show that the proposed method did not outperform existing baselines. The main claim of the paper (benefit of two-stage procedure) is not supported by ablation study.

---

### Official Review · Reviewer_g92F · 2021-11-04

**Correctness:** 3
**Technical Novelty And Significance:** 2
**Empirical Novelty And Significance:** 1
**Recommendation:** 3
**Confidence:** 4

**Main Review:**


Strength:
1. The whole framework well adapts existing methods with good motivations.
2. The proposed model outperforms other baselines in the PlogP optimization benchmark.


Weakness:
1. The novelty is limited. They apply an existing genetic algorithm framework and the fitness functions are straightforward. The operations are the most novel point in this paper, however, its novelty is very limited because its modification is not significant compared to the existing one. In addition, the graph-based crossover is an adaptation of GB-GA.
2. Experiments only show one property optimization, but they claim it as a multi-objective problem. There are many molecular properties that need to be optimized at the same time, which should have been considered in this paper in order to prove its superiority. Please refer to the multi-objective optimization benchmark in CMG[1].
3. Only PlogP optimization was shown in the experiment. There are other properties for the single objective optimization such as QED or DRD2.

[1] Shin, Bonggun, et al. "Controlled molecule generator for optimizing multiple chemical properties." Proceedings of the Conference on Health, Inference, and Learning. 2021.

**Summary Of The Paper:**

This paper presents a genetic algorithm framework for molecular optimization. It produces valid molecules through the crossover and mutation operations along with appropriate fitness functions. The evaluation was done in one of the property optimization tasks.


**Summary Of The Review:**

The reason why I can't recommend this as one of the ICLR papers is
1. The technical novelty is almost none.
2. Many other experiment results should be provided in order to assess various aspects of the proposed model.

---

### Decision · Program_Chairs · 2022-01-20

**Decision:**

Reject

**Comment:**

The paper describes a genetic algorithm for molecular optimization under constraints. The aim is to generate molecules with better properties while close to an initial lead molecule. The proposed approach is a two-stage one. The first stage aims to satisfy constraints and searches for feasible molecules that are similar to the lead. The second stage optimizes the molecular property. The method is evaluated on logP optimization task, with minor improvement over previous work.

The reviewers point out the following strengths and weaknesses:

Strengths:

- Molecular optimization under structural constraints is an important research direction.
- Comprehensive related work section.

Weaknesses:

- Lack of novelty because it is a standard application of genetic algorithm.
- The results show that the proposed method did not outperform existing baselines.
- The main claim of the paper (benefit of two-stage procedure) is not supported by ablation study.
- The authors only conduct experiments on improving LogP, which is a benchmark that is too easy and not challenging.
- The objective function and cross-over operation are the same or very similar to previous work.
- The experimental evaluation is limited, and the overall setting is not very relevant to real-world tasks.

Overall, all reviewers vote for rejection. It is clear that the paper needs more work before it can be published.